# The Natural History of Hepatitis C Virus Infection and Disease in the Era of Curative Therapy with Direct-Acting Antivirals

**DOI:** 10.3390/v17030319

**Published:** 2025-02-26

**Authors:** Maurizia Rossana Brunetto, Ferruccio Bonino

**Affiliations:** 1Hepatology Unit and Laboratory of Molecular Genetics and Pathology of Hepatitis Viruses, Reference Centre of the Tuscany Region for Chronic Liver Disease and Cancer, University Hospital of Pisa, Via Paradisa 2, 56124 Pisa, Italy; maurizia.brunetto@unipi.it; 2Institute of Biostructure and Bioimaging, National Research Council, Via De Amicis 95, 80145 Naples, Italy; 3Department of Clinical and Experimental Medicine, University of Pisa, Pisa University Hospital, Via Roma 55, 56126 Pisa, Italy

**Keywords:** HCV, hepatitis C, natural history, antiviral therapy, HCV vaccine, HCV elimination

## Abstract

The availability of highly effective direct-acting antivirals (DAAs) that cure individuals infected with HCV has changed completely the natural history of HCV infection and chronic hepatitis C. In sustained responders to DAAs, the most common clinical-pathologic outcome has become liver disease regression, cirrhosis re-compensation, and the de-listing of transplant candidates. However, careful scrutiny of liver disease cofactors and outcome predictors in treated patients is mandatory for an appropriate personalized surveillance of the residual risk for hepatocellular carcinoma. Since successful treatment with DAAs does not confer protective immunity against HCV reinfection, an effective vaccine is critically needed to control HCV infection. Meanwhile, it is mandatory to enhance universal access to DAAs, to test asymptomatic high-risk groups who are the main source of transmission, and to screen people who inject drugs (PWID), men who have sex with men (MSM), and sex workers, and to assure safe medical procedures with the provision of disposable needle and syringes.

## 1. Introduction

In 2015, the unprecedented capacity of curing hepatitis C Virus (HCV) infection and disease by direct-acting antivirals (DAAs) prompted the World Health Organization (WHO) to propose the global target of eliminating the public health threat of HCV by 2030 [1]. The HCV elimination plan projected a 90% reduction in new HCV infections and a 65% reduction in deaths globally by 2030 [2]. Some 10 years later, we are far from making HCV elimination truly sustainable, and reports now project reaching this goal by 2050 and suggest actions to improve both HCV screening and treatment [3]. Approximately 1.75 million new cases of HCV infection were reported worldwide in 2015, and in 2019, globally, 57.8 million (0.8% of total) people are estimated to be living with chronic HCV infection [4]. Among people with HCV infection, 15.2 million (95% CI 12.1–19.0) had been diagnosed between 2015 and 2019, and 9.4 million (7.5–11.7) people diagnosed with hepatitis C infection were treated with direct-acting antiviral drugs between 2015 and 2019 [4]. The number of reported acute and chronic HCV hepatitis cases declined in the USA from 2015 to 2020 (average decline, −13,177 per year) [5]. However, the proportion of hepatitis C cases among those aged 18–39 years increased by an average of 1.4% per year, whereas among individuals aged 40–59 years, it decreased by an average of 2.3% per year [5]. The current geographical distribution of HCV infection varies widely worldwide, and several factors such as socio-economic conditions, health care organization (both screening and treatment), and customary practices contribute to the important regional disparities in incidence rates. In this short review, we address the most relevant issues that are currently influencing the natural history of HCV infection.

## 2. Source of Infection

After the discovery of the hepatitis C virus (HCV), the search for homologous animal viruses (genus *Hepacivirus*) began, as did the study of their origin, which is estimated at approximately 22 million years ago. *Hepacivirus* diversity stems from evolutionary dynamics that have occurred over time and were influenced by cross-species transmission [6]. Recently, the importance of rodents as *hepacivirus* hosts has been emphasized as models for investigating HCV infection dynamics, though there is no evidence yet of any zoonotic origin of HCV [6,7]. Given an effective worldwide control of blood donors and derivates, the major modes of HCV transmission remain iatrogenic infection (nosocomial exposure to unsafe medical procedures) in low to intermediate income countries, and needlestick in high-risk groups of persons who inject drugs (PWID (mainly if without access to needle and syringe provision), men who have sex with men (MSM), and prisoners) in high-income countries [8].

### Reinfection

In individuals highly exposed to HCV, such as PWID, reinfection is common, and in those who are reinfected a small proportion clear the virus, suggesting that natural protection is possible. However, the majority develop a persistent infection, indicating that a natural sterilizing immunity is difficult to achieve. Nevertheless, the evidence that individuals can spontaneously clear HCV infection suggests that a protective response involves both humoral and cellular immune responses, and thus any successful vaccine will need to trigger both these types of response. A broad HCV envelope 2 (E2)-specific memory B cell response is critical for protection against antigenically diverse HCV variants [9], and the hunt for an effective HCV vaccine is currently underway [10]. However, since there is no effective vaccine available yet and successful treatment with DAAs does not confer protective immunity, HCV reinfection after sustained response to antivirals can occur, undermining significantly the efforts to eliminate HCV. Many studies assessing the changes of HCV reinfection incidence among PWID- and HIV-infected people following the introduction of DAA have been performed [11,12,13,14,15]. The evidence shows that the proportion of incident HCV cases due to reinfection was highest during periods of broad access to direct-acting antivirals, which highlights the importance of continuing to reduce ongoing risks and of testing people at risk [16].

## 3. HCV-Induced Disease Liver Disease

HCV infection can cause either acute or chronic hepatitis. However, more than 30 years after the introduction in 1991 of routine blood screening for HCV antibody and further improvements in testing, transfusion-related acute hepatitis has significantly declined [5]. Nowadays, newly infected individuals are usually asymptomatic, and 15–45% (approximately one-third) of them resolve, while 55–85% (about two-thirds) progress to chronic HCV infection, defined as the persistence of HCV-RNA in the blood for more than 6 months, the same arbitrary time cut-off used to defined chronic hepatitis B virus infection. At variance with HBV, chronic HCV infection almost invariably causes chronic hepatitis C (CHC) in all infected individuals. Thus, chronic HCV infection and CHC can be used as synonyms and still represent a serious health care burden worldwide (WHO 9 April 2024. https://www.who.int/news-room/fact-sheets/detail/hepatitis-c). HCV infection remains at one of the highest rates in China, where a slow upward tendency was still reported from 2012 to 2017, and only now appears to be declining [17].

### 3.1. The Impact of DAAs

HCV natural history studies are limited by not knowing the time of infection. By contrast, studies of the impact of DAAs provide consistent evidence for registering clinical outcomes within a known period, from the time of starting treatment and comparing treated patients with or without a sustained virologic response (SVR). Several studies analyzed the annual cumulative incident events and incidence rate/1000 person-years of follow-up for liver cirrhosis, hepatic decompensation, hepatocellular carcinoma (HCC), and mortality rates [18,19,20,21,22,23,24,25,26,27]. The effectiveness of direct-acting antivirals (DAAs) in eradicating HCV is well-established and represents one of the greatest achievements of medical therapeutics. Chronic HCV is associated with a significantly higher risk of mortality, and sustained virologic response (SVR) from DAAs was associated with a significant reduction in the risk of all-cause, liver- and drug-related mortality. Unequivocal short- and long-term clinical benefits for treated patients are both hepatic (e.g., improvement of liver function tests, regression of fibrosis and cirrhosis, reduction in risk of HCC and liver-related mortality) and extrahepatic (e.g., improvement in risk and control of, cryoglobulinemia, lymphoma, health-related quality of life, and, possibly, overall mortality) [18]. In a large general-population Canadian cohort of approximately 10,000 people with more than 5 years of follow-up, the effect on mortality risk of DAA-induced SVR compared to no-treatment showed an overall 81% reduction in mortality rate, a 78% reduction in liver-related deaths, and a 74% reduction in drug-related mortality risk [19]. In addition, in the treated group, SVR compared to no-SVR was associated with an 81% reduction in all-cause deaths and a slightly higher reduction in liver-related mortality (87%), but a lower (64%) reduction in drug-related mortality risk. The reduction in mortality risk was lower among older patients with cirrhosis at treatment initiation, while older age and problematic alcohol use were significantly associated with a higher risk of liver-related mortality. Thus, DAAs treatment before the development of cirrhosis is necessary to substantially reduce the risk of liver-related mortality. Similar results from many other studies prompted many developed countries to remove fibrosis-based restrictions on treatment coverage. However, treatment access needs to be scaled-up quickly to achieve the full benefits of DAAs in reducing mortality. Overall, SVR delivers, even in advanced stages of disease, significant benefits, with a reduction in the risk of developing HCC (−70%) and overall liver-related mortality and/or liver transplant (−90%) [21,23]. Treating HCV with DAAs greatly reduced deaths related to extrahepatic manifestations, including cardiovascular, cerebrovascular, and chronic kidney diseases, as well as diabetes, mood and anxiety disorders, and other mental health conditions, with an overall reduction in mortality risk ranging from 78 to 84% [18,19,20,25,26]. HCV is associated with a higher risk of metabolic diseases such as type 2 diabetes (T2D), whose risk is significantly reduced in case of SVR as well as that of renal and cardiovascular complications [24]. A large prospective study showed a 52% reduction in the overall mortality rate and a 34% reduction in the incidence of HCC [26]. Furthermore, a recent report on a large population study in British Columbia showed that treating HCV infection with DAAs was associated with significant reductions in deaths related to extrahepatic manifestations, including cardiovascular, cerebrovascular, and chronic kidney diseases, diabetes, mood and anxiety disorders, and mental health conditions, with the reduction in mortality risk ranging from 78 to 84% [27]. These findings emphasize the critical importance of timely diagnosis and treatment of HCV to prevent deaths associated with both hepatic and extrahepatic causes [20,27].

### 3.2. Natural History Shift Induced by DAAs

The effectiveness of DAAs therapy that includes the removal of the primary aetiologic factor of liver damage even in patients with advanced liver disease at the time of therapy initiation have changed completely the natural history of CHC that was usually conceptualized in the unidirectional concept of disease progression. 



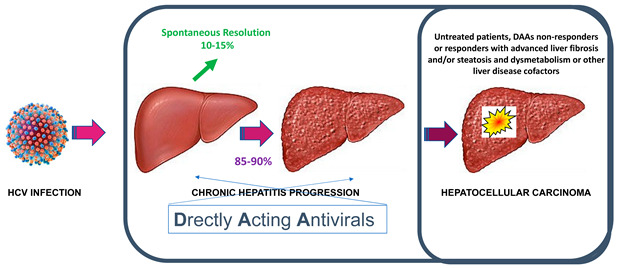



In the era of DAAs, given the high rate of sustained response to antiviral therapy, the most common trajectory of the clinic-pathologic outcome has become that of liver disease regression and cirrhosis re-compensation that can even enable the de-listing of transplant candidates. Thus, new efforts are being made to generate consensus on defining the new concepts of liver disease regression and cirrhosis re-compensation [28,29]. Future studies are required to shed new light on the natural history of hepatic re-compensation and the molecular mechanisms of disease regression, including the remodeling of the cirrhotic lesions at histology, as well as to assess modifying factors and potential non-invasive biomarkers.

## 4. Factors and Predictors of Clinical Outcomes

HCV hijacks many host metabolic processes in an effort to aid its replication, and the resulting hepatic metabolic dysfunction underpins many hepatic and extrahepatic manifestations of HCV infection. Consequently, the natural history of CHC is substantially influenced by the host metabolic status: hepatic steatosis, obesity, insulin resistance/type 2 diabetes are major determinants of CHC progression toward HCC [30]. A study of a large cohort of DAAs-treated patients with CHC and cirrhosis showed that a genetic risk score (GRS) combining hepatic fat accumulation with variants in PNPLA3 (patatin-like phospholipase domain containing 3), MBOAT7 (membrane bound O-acyltransferase domain containing 7), TM6SF2 (transmembrane 6 superfamily member 2) and GCKR (glucokinase regulator) associated with HCC development independently of classical risk factors, including liver disease severity [31]. This and other predictive models based on non-invasive markers [32,33,34] suggest that combinations of clinical and genetic predictors may improve HCC risk stratification in patients with advanced liver disease who undergo DAAs treatment. Of course, the evidence that different DAA regimens in children induce higher cure rates with minimal side-effects and a shorter duration of therapy confirms that the earlier the treatment the better clinical outcome [35].

In clinical practice, it is very important to take into account all individual risk factors for an appropriate tailoring of prevention and surveillance [36]. Since insulin resistance or type 2 Diabetes (T2D) are known to increase HCC among individuals with CHC, anti-diabetic and obesity therapies are highly recommended for individuals with metabolic syndromes after the eradication of HCV in order to reduce the risk of HCC. A recent study evaluated in a large population of 7249 individuals whether metformin reduced HCC risk among individuals with CHC and T2D after a sustained viral response (SVR) to DAAs therapy. Metformin was shown to greatly reduce the risk of HCC and all liver-related complications in individuals with SVR [37]. A periodic ultrasound-based surveillance for hepatic lesion is important in patients with advanced fibrosis/cirrhosis who achieved SVR after DAAs [38]. However, since both fibrosis indices and stages are major risk predictors [34,39], the non-invasive liver stiffness measure by transient elastography (TE) is very useful to monitor in the single patient the effective decline of TE from baseline to end-of-therapy and during the follow-up, since the lack of normalization of liver stiffness after SVR beckons the persistence of morphological alterations to the liver structure [40,41].

## 5. Conclusions

From the evidence of the natural history of HCV infection, the efforts to slow the infection rate in the absence of a vaccine against HCV rely mostly on anti-HCV testing in routine medical examinations and active surveillance of high-risk populations, such as PWID, MSM, and sex workers. The target populations for hepatitis C control—by assuring safe medical procedures and/or providing needles and syringes—are the elderly, sexually active people, PWID, migrants, and renal dialysis patients. Asymptomatic people remain the main source of transmission. Thus, for the better prevention and control of HCV, it is very important to improve the screening of asymptomatic people in high-risk groups, identifying secondary HCV infections, especially in human immunodeficiency virus (HIV) co-infected individuals. Furthermore, it is mandatory to enhance universal access to effective DAAs treatment to improve the cure rate of HCV infection. Thus, at least in countries with higher rates of infection, regional-specific surveillance with sentinel sites should be implemented to monitor HCV infection. In addition, informing both treatment providers and patients about the overall benefits of DAAs therapy is mandatory in promoting timely HCV treatment, and earlier treatments warrant better clinical outcomes. Despite the availability of highly effective direct-acting antivirals (DAAs) that cure individuals infected with HCV, developing a vaccine is critically needed for achieving HCV elimination. In patients achieving SVR after DAAs, careful scrutiny of liver disease cofactors and outcome predictors is mandatory for the appropriate personalized surveillance of individuals with residual HCC risks.

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
