# Peer review of "The Natural History of Hepatitis C Virus Infection and Disease in the Era of Curative Therapy with Direct-Acting Antivirals"

_viruses, 2025, doi:10.3390/v17030319_

Round 1
Reviewer 1 Report
Comments and Suggestions for Authors
I would like to express my gratitude for the opportunity to review this manuscript. The topic is highly relevant and timely, given the transformative impact of DAAs on the natural history of HCV infection and chronic hepatitis C. The discussion on the regression of liver disease, cirrhosis re-compensation, and delisting of transplant candidates highlights the substantial clinical benefits of these therapies. Moreover, the emphasis on the necessity for continued surveillance to assess HCC risk and the call for enhanced prevention strategies, including a vaccine, make this paper an important contribution to the field.
However, I have a few comments and suggestions for improvement:
- References:
- Please verify the accuracy of the references, as it appears that references 2 and 42 are identical.
- Additionally, the citations in lines 31, 60, and 89 seem to be incorrectly referenced.
- Abbreviations:
- The abbreviations used in the abstract, such as PWID and MSM, should be expanded upon first mention to improve readability for a broader audience.
- It would also be beneficial to review abbreviations throughout the main text to make sure they are expanded only at their first appearance (e.g., SVR, DAA, and HCC).
Author Response
According to the very helpful suggestions of Reviewer 1 we corrected the mistake of having inserted twice reference 2 also in position 42 instead of the correct reference (EASL Guideline publication the following : Clinical Practice Guidelines on non-invasive tests for evaluation of liver disease severity and prognosis – 2021 update. J Hepatol. 2021 Sep 1;75(3):659–89) that has been substituted to the previous one. We we checked the references at lines 31, 60 and 89 and they are correct indeed. We agree on the suggested way to present abbreviations at their first appearance and thus we aligned all the abbreviations to be expanded upon first mention along all the manuscript .
Reviewer 2 Report
Comments and Suggestions for Authors
In this review, the authors discussed source of HCV infection, reinfection, mechanisms of HCV-induced liver diseases, the use of direct acting antiviral drugs in HCV-infected patients, and factors affecting HCV-clinical outcomes.
The review is short and could not meet with the requirement of review.
Major concerns
1- The degree of plagarism is very high and not acceptable as reported by MDPI report.
2- Several reviews had published the same topic before such as
https://academic.oup.com/cid/article/77/Supplement_3/S238/7242429
https://www.sciencedirect.com/science/article/pii/S016344531300251X
https://link.springer.com/article/10.1007/s40471-017-0108-x
https://www.mdpi.com/2077-0383/12/6/2195
Author Response
We agree that the review issue that was given to us by the editor of the Special Issue was objective of many reviews already and this is not surprising in the current era of over-publication. Nevertheless I could not refuse the kind invitation that was motivated by the fact that we were the clinicians in charge of the worldwide collection of sera to confirm the discovery of HCV in the original papers of Nobel Price winner, Michael Houghton (1989). Coauthor-ship of these manuscript document this. Certainly we did not present works or ideas from another source as our own without full acknowledgement in the references as «plagiarism » would mean . Of course a review is not meant to index all reviews already published on the same issue and particularly the least relevant. Actually the reviewer appears ignorant not only of the meaning of the word «plagiarism » but also of its writing since he wrote « plagarism ».
Reviewer 3 Report
Comments and Suggestions for Authors
The manuscript provides a comprehensive overview of HCV infection, its natural history, and the impact of direct-acting antivirals (DAAs). It integrates up-to-date epidemiological data and discusses global efforts to eliminate HCV. The focus is on predictors of clinical outcomes and the shift in disease progression due to DAAs.
The manuscript heavily focuses on DAAs and their clinical impact but does not explore the limitations in real-world applications, such as accessibility in low-income countries, treatment resistance, or patient compliance. While the discussion on predictors of clinical outcomes is strong, genetic factors influencing treatment response could be expanded. The mechanisms underlying metabolic alterations in HCV-infected individuals should be elaborated to provide a more mechanistic understanding.
The manuscript is written in formal academic English, maintaining a professional tone. Scientific terminology is used accurately and appropriately. Citations are well-integrated and support key points.
In section 3.1 you discuss the impact of DAAs on improvements in extrahepatic conditions such as metabolic disorders after HCV clearance. The impact of DAAs extends beyond viral eradication, as successful treatment has been associated with significant metabolic improvements, including better glycemic control and reduced risk of diabetes-related complications (see DOI 10.31925/farmacia.2022.4.9).
In section 4 you discuss factors and predictors of clinical outcomes like host metabolic factors, insulin resistance, fibrosis scores, and their impact on treatment response. Previous studies have demonstrated that treatment failure in CHC is linked to factors such as advanced age, high HCV-RNA viral load, and metabolic disturbances, including insulin resistance, elevated fasting glucose, and high homeostatic model assessment of insulin resistance (HOMA-IR) index, alongside liver histology features like fibrosis and steatosis (see DOI 10.47162/RJME.61.4.20). These parameters can help predict early virological response (EVR) and sustained virological response (SVR), thus influencing therapeutic strategies.
Author Response
Comments of Reviewer 3 just emphasize the consistency of our report referring also to minor publications which are not fully pertinent with the major issue already discussed in the review.